# Impact of Early Nutritional Intervention During Cancer Treatment on Dietary Intakes and Cardiometabolic Health in Children and Adolescents

**DOI:** 10.3390/cancers17010157

**Published:** 2025-01-06

**Authors:** Josianne Delorme, Andra Dima, Véronique Bélanger, Mélanie Napartuk, Isabelle Bouchard, Caroline Meloche, Daniel Curnier, Serge Sultan, Caroline Laverdière, Daniel Sinnett, Valérie Marcil

**Affiliations:** 1Centre de Recherche Azrieli du CHU Sainte-Justine, Montreal, QC H3T 1C5, Canada; josianne.delorme@umontreal.ca (J.D.); andra.daniela.dima@umontreal.ca (A.D.); veronique.belanger.7@umontreal.ca (V.B.); melanie.napartuk@umontreal.ca (M.N.); isabelle.bouchard.hsj@ssss.gouv.qc.ca (I.B.); caroline.meloche.cnmtl@ssss.gouv.qc.ca (C.M.); daniel.curnier@umontreal.ca (D.C.); caroline.laverdiere@umontreal.ca (C.L.); daniel.sinnett@umontreal.ca (D.S.); 2Department of Nutrition, Faculty of Medicine, Université de Montreal, Montreal, QC H3T 1A8, Canada; 3School of Kinesiology and Physical Activity Sciences, Université de Montréal, Montreal, QC H3T 1J4, Canada; 4Department of Psychology, Université de Montréal, Montreal, QC H2V 2S9, Canada; 5Service of Hematology-Oncology, CHU Sainte-Justine, Montreal, QC H3T 1C5, Canada; 6Department of Pediatrics, Faculty of Medicine, Université de Montréal, Montreal, QC H3T 1C5, Canada

**Keywords:** children, adolescent, cancer, nutrition, dietary intakes, pediatric oncology, intervention, cardiometabolic health

## Abstract

Pediatric cancer survivors have a higher risk of developing cardiometabolic complications compared to their peers. This study evaluates the impact of VIE (Valorization, Implication, Education), which is a personalized intervention program that integrates nutrition, physical activity, and psychological support, on dietary intake and cardiometabolic health among children and adolescents following cancer treatment. The participants exposed to the VIE intervention showed specific improved dietary intakes at an average of 1.3 years post-treatment compared to a control group; this was reflected by lower caloric and higher calcium intakes. Furthermore, the participants who were highly involved in the nutritional intervention had greater protein-derived energy intake than the controls. Only the adolescents in the intervention group exhibited a trend toward a lower proportion of cardiometabolic risk factors. While our findings show limited clinical impact, they underscore the importance of exploring additional strategies to improve the diet of all pediatric cancer patients during and after treatment and show the need for further research to assess the long-term impact of such interventions on health.

## 1. Introduction

Cancer is the leading cause of disease-related death in children, with approximately 1000 new cases diagnosed annually in Canada [1,2]. Although advances in treatment have improved pediatric cancer survival rates to 84% [3], chemotherapy and radiotherapy often result in acute side effects, including dysgeusia, mucositis, taste and smell dysfunction, nausea, and other gastrointestinal symptoms [4]. These side effects frequently lead children and adolescents to consume low-quality foods, negatively affecting dietary intake and nutritional status [4,5,6,7,8,9,10,11]. Furthermore, acute cardiometabolic complications are also frequently observed during pediatric cancer treatment [12,13]. For instance, insulin resistance occurs in up to 50% of acute lymphoblastic leukemia (ALL) patients during treatment [14]. Obesity also poses challenges throughout treatment [15], with prevalence increasing from 15% at diagnosis [16] to 21% post-treatment [17]. Additionally, obesity at diagnosis is associated with a higher risk of relapse in pediatric ALL patients over the age of 10 [18].

When considering the evolution of health after treatment, childhood cancer survivors have a higher risk of chronic health conditions compared to their peers [19,20,21,22,23], with two-thirds developing at least one chronic illness during their lifetime [24]. Previously, we reported that, within an average of 1.4 ± 0.8 years (y) following cancer treatment completion, 35% of children and adolescents exhibited dyslipidemia, 26.3% had high blood pressure, 11.5% were obese, and 8.1% presented with prediabetes [25]. Long-term studies underscore the persistence of cardiometabolic risk as, for instance, blood lipid abnormalities affected up to 50% of childhood ALL survivors in a cohort of 80 participants with a mean age of 21.1 ± 0.8 y, who were assessed after a mean follow-up time of 12.4 y since cancer diagnosis [26]. Other studies among childhood ALL survivors reported that hypertension affected up to 46.4% and obesity affected 44.9%; 31.4% exhibited fasting hyperglycemia or required treatment at long-term assessment, with time since diagnosis ranging from 5.2 to 26.1 y [23,27,28,29]. Furthermore, inadequate calcium and vitamin D intake during and after treatment are particularly common, increasing the risk of bone complications in long-term pediatric cancer survivors [30,31,32,33,34].

The mechanisms underlying the onset of cardiometabolic complications during and after pediatric cancer treatment remain poorly understood. However, extensive evidence from the general population highlights the critical role of diet and lifestyle in promoting cardiometabolic health [35,36,37,38,39,40]. In parallel, studies show that survivors of pediatric cancer often maintain unhealthy dietary habits acquired during treatment, such as a preference for fast food and inadequate micronutrient intake, further compounding the risk of long-term cardiometabolic complications [7,8,30,31,32,33,34]. Typically, lifestyle interventions during childhood cancer have been implemented late in treatment or post-recovery [41,42,43,44,45,46], limiting their preventive potential. Given that childhood and adolescence are pivotal for establishing lifelong dietary habits, early intervention is crucial to mitigate the adoption and persistence of unhealthy eating behaviors. Accordingly, the VIE study (Valorization, Implication, Education) is a multidisciplinary initiative implemented shortly after diagnosis to promote healthy lifestyle habits among pediatric cancer patients and their families. It provides education and support in nutrition, psychology, and physical activity to manage treatment-associated side effects and complications [25,47,48,49,50,51].

The aim of this study is to assess the impact of the VIE intervention on nutritional intakes and the cardiometabolic health of children and adolescents after the end of their treatment (i.e., 1.3 y after cancer treatment, on average). The main objective is to compare diet and cardiometabolic health outcomes between the VIE intervention group, including a subgroup of participants who were highly involved in the nutritional intervention, and a control group who received standard medical care. It also aims to assess the impact of the intervention based on age at diagnosis (children and adolescents).

## 2. Materials and Methods

### 2.1. Study Design and Ethics

The VIE study was conducted at the Centre Hospitalier Universitaire Sainte-Justine (CHUSJ) in Montreal, Quebec, Canada from June 2017 to August 2022. Detailed descriptions of the nutritional, physical activity, and psychosocial components of the intervention have previously been described [25,47,48,49,50,51]. This study used a two-group design: an intervention group (participants exposed to the VIE intervention) and a control group (participants treated prior to the implementation of the VIE intervention), both of which correspond to convenience samples derived from the number of patients diagnosed, treated, and followed up for cancer within the institution. All the study procedures were approved by the Ethics Review Board of the CHUSJ (#2017–1413) and conducted in accordance with the Declaration of Helsinki. Written informed consent was obtained from all the participants and/or from their parent/legal guardian where applicable.

### 2.2. Recruitment of Participants

#### 2.2.1. Intervention Group

Participants were recruited between February 2018 and December 2019. The eligibility criteria were: (1) age of 21 y old or younger at diagnosis; (2) being treated with chemotherapy or radiotherapy; (3) being able to give informed consent (by parents or legal guardians); and (4) diagnosis received less than 12 weeks prior to the time of recruitment. The medical team could withdraw participants from the study if their health condition no longer allowed them to participate (e.g., transitioned to palliative care).

#### 2.2.2. Control Group

Participants were recruited from June 2017 to October 2019. The eligibility criteria were: (1) age of 21 y old or younger at diagnosis; (2) had been treated with chemotherapy or radiotherapy; and (3) had completed cancer therapy before the VIE program was initiated. The exclusion criterion was the experience of relapse or a second neoplasm. The participants that were included in the analysis had to have completed their cancer treatment before the VIE intervention was implemented and, as such, to have received standard medical care.

### 2.3. Components of the VIE Intervention

#### 2.3.1. Nutritional Intervention

The details and feasibility of the VIE nutritional intervention have previously been described [47,49]. In sum, the intervention was designed to promote healthy eating behaviors, support optimal growth and development, improve abnormal growth patterns during and after treatment, and ultimately prevent cardiometabolic complications. The approach utilized motivational interviewing to support improvement in the participants’ diets, while guiding parents in accompanying their children through lifestyle changes and emerging challenges. Two registered dietitians (RD) conducted the intervention, which included: (1) an initial assessment; (2) follow-up visits planned every 2 months over one year; and (3) a one-year assessment. After the first year of nutritional intervention, the necessity for continued follow-up was discussed with the participant and/or their parents. When additional follow-ups were needed, the frequency of appointments was adapted to meet the participant’s preferences. As previously reported [47], levels of participation and engagement with the nutritional intervention varied among participants. The participation rate was calculated for each participant as the number of completed follow-up visits out of six planned visits, categorized as low (0–1 visit), moderate (2–3 visits), or high (≥4 visits). The engagement levels of the participants were categorized as either low, moderate, or high according to the RD’s subjective assessment made after one year of intervention, which was based on the ease of scheduling appointments and the global interest towards the intervention [47]. Here, high involvement in the nutritional intervention was defined as the demonstration of both high participation (≥4 visits) and high engagement level.

#### 2.3.2. Physical Activity Intervention

The physical activity intervention aimed to support the long-term well-being of participants by encouraging the adoption of an active lifestyle during and after cancer treatment [48]. The intervention was delivered by kinesiologists and consisted of a baseline assessment to evaluate physical capacity, followed by an individualized training program tailored to the participant’s needs, abilities, personal goals, medical condition, and functional status. Over a six-week period, the participants engaged in two to three 45-min physical activity sessions per week, as described in previous work [48]. These sessions could be conducted either in the hospital or remotely via an online platform, under the guidance of a kinesiologist in both cases. Thirty minutes of each session were dedicated to moderate-intensity aerobic exercise. The remaining 15 min were allocated to adapted functional training, including resistance, flexibility, and balance. To ensure that the program remained adapted to the participant’s medical condition and needs, a reassessment was scheduled every two months, or sooner if necessary.

#### 2.3.3. Psychosocial Intervention

The psychosocial intervention was offered to the parents of the participants in the VIE intervention group. The proposed support, named “Taking Back Control Together”, has been described in detail [52]. In sum, the intervention was conducted by a trained psychologist and aimed at enhancing problem-solving skills in parents through a series of six 60-min sessions spread over a period of 6 to 8 weeks. The first four sessions were conducted individually (i.e., with a single parent) to address personal strategies, while the last two involved the participation of both parents, focusing on couple communication and dyadic adjustment. A study on the feasibility and exploratory effects of the intervention on parental emotional distress has recently been published [51].

### 2.4. Data Collection

Between January 2021 and August 2022, an end-of-study (EOS) assessment was conducted for the intervention group. This assessment included the collection of clinical, nutritional, anthropometrical, and biochemical data. To enable robust comparisons, these data were also collected for the control group, during a single visit between June 2017 and October 2019.

#### 2.4.1. Demographic, Diagnostic, and Treatment-Related Data

Data were also gathered from medical charts, such as demographic data, which included sex and date of birth to determine age at diagnosis, age at the end of treatment (EOT), and age at EOS assessment. Data on diagnosis were also collected to determine the cancer type (e.g., leukemia, lymphoma, and sarcoma). Finally, the treatment-related data included exposure to radiotherapy and/or corticosteroid therapy and date of last treatment, which was used to calculate treatment duration and time since EOT.

#### 2.4.2. Nutritional Data

Dietary data were based on reported energy intake collected using a 24-h recall administrated by the RD. Nutritional data were analyzed using Nutrific^®^ software developed by the Department of Food Science and Nutrition of Laval University [53] and based on the 2015 Canadian Nutrient File [54]. Total daily energy (kcal) was calculated and expressed per body weight (kcal/kg) and as a percentage of the estimated energy requirement (EER). The calculations were performed using guidelines from the National Research Council’s Recommended Dietary Allowances (10th Edition) [55] and from the updated recommendations on nutrition and hydration requirements from StatPearls [56].

Assessment of dietary intake encompassed four macronutrients, four micronutrients, and two diet quality scores, all derived from 24-h food recall. Macronutrient intake, including fat, carbohydrates, and protein, was reported as g/kg, % of total energy [TE], and % of recommended dietary allowance [RDA] for protein. Dietary fiber intake was described as g/1000 kcal and % of adequate intake [AI]. Assessment of micronutrient intake included sodium as mg/1000 kcal and %tolerable upper intake level [UL], calcium and vitamin C as mg/1000 kcal and %RDA, and vitamin D as μg/1000 kcal and %RDA. As previously described [49], overall diet quality was evaluated using the diet quality index (DQI), which is a continuous scale ranging up to a maximum of 100 points [57], and the healthy diet indicator (HDI), which categorizes adherence to nine nutritional recommendations as low, medium, or good [58,59,60].

#### 2.4.3. Anthropometric Data

Anthropometric measurements and the z-score calculation were performed with the participants in the control and intervention groups as previously described [25,49]. Following the same method, measures were reconducted for the intervention participants at the EOS assessment. Briefly, waist circumference (WC) and mid-upper arm circumference (MUAC) were assessed using a non-stretchable measuring tape with an accuracy of 0.1 cm. The tricipital (TSFT) and subscapular (SSFT) skin folds were collected using a Harpenden skin fold caliper with a precision of 0.2 mm from the average of 2 consecutive measurements. The oncology nurse measured the weight and height of the participants to the nearest 0.1 kg and 0.1 cm, and these measurements were used to compute body mass index (BMI (if >2 y old) or the weight-for-length (W-L) ratio (if <2 y old) Finally, the Microsoft^®^ Office Excel^®^ tool (developed by the British Columbia Children Hospital) and the Canadian Pediatric Endocrine Group (based on the 2014 version of Growth Charts for Canada) were used to calculate the BMI and W-L z-scores [61], while the WC, MUAC, TSFT, and SSFT z-scores were derived from populational reference data [62,63,64].

#### 2.4.4. Biochemical Assessment and Cardiometabolic Risk Factors

The day of the EOS assessment, blood pressure (BP) and biochemical parameters (total cholesterol [TC], low-density lipoprotein cholesterol [LDL-C], high-density lipoprotein cholesterol [HDL-C], glycated hemoglobin [HbA1c], and 25-hydroxyvitamin D [25(OH)D]) were gathered from medical records and from blood sample collection, respectively. The presence of cardiometabolic risk factors, namely high BP, blood lipid abnormalities, high HbA1c, and obesity, was further determined for each of the participants using the same criteria previously used for VIE participants [25,49]. Briefly, high BP, including both systolic and diastolic measurements, was assessed based on guidelines from the American Academy of Pediatrics [65], while the z-scores for sex, age, and height were calculated using the British Colombia Children’s Hospital Microsoft^®^ Office Excel^®^ tool (version: 2022/01/17). Blood lipid abnormalities, which represent the presence of high TC, high LDL-C and/or low HDL-C, high HbA1c, and obesity, were determined using populational established criteria based on sex and age, as presented in Appendix A [65,66,67,68,69,70]. Finally, 25(OH)D levels were measured in serum using liquid chromatography–tandem mass spectrometry to determine vitamin D status and were classified according to the 2016 Consensus Report of Experts, where sufficiency is defined as values > 50 nmol/L, insufficiency as 30–50 nmol/L, and deficiency as <30 nmol/L [71].

### 2.5. Data Analysis

The continuous variables are reported as mean ± standard deviation (SD) and median with a range (min–max), while the categorical data are presented as the proportion of participants per group (%). The Pearson chi-square test, or its alternative the Fisher exact test if more than 25% of the expected occurrences were <5 and if any expected occurrence was null, was used to assess the relationship between groups (intervention vs. controls and intervention subgroup vs. controls) and the categorical variables, including the participants’ socio-demographic characteristics (sex [female vs. male], cancer diagnosis [leukemia vs. lymphoma vs. sarcoma vs. other]), the classification of intake of fat and carbohydrates (>acceptable macronutrient distribution range [AMDR] vs. = AMDR vs. <AMDR), protein (>RDA vs. <RDA), total dietary fiber (>AI vs. <AI), sodium (>UL vs. <UL), as well as calcium, vitamin C and vitamin D (>RDA vs. <RDA), the classification of HDI scores (low vs. medium vs. high), the classification of biochemical values (TC, LDL-C, HDL-C, and vitamin D status), and the cardiometabolic risk factors (high BP, high HbA1c, blood lipid abnormalities, obesity, ≥2 risk factors). Two-tailed Student’s *t*-tests were used to compare differences in the means of the continuous variables [age, weight, height, BMI, MUAC, TSFT, SSFT, dietary intake, diet quality scores, and biochemical profile] between the control and the intervention groups. The relationship between sampling season (summer vs. fall vs. spring vs. winter) and the vitamin D status (sufficiency vs. insufficiency vs. deficiency) of the participants was assessed using the chi-square test. The groups were stratified according to age at diagnosis, and the participants were classified as either children (<10 y) or adolescents (≥10 y). Analyses were conducted using the same statistical tests as described above. Analyses were performed using SPSS version 25.0 (IBM, Armonk, New York) and R version 4.4.1 (R Foundation for Statistical Computing, Vienna, Austria) [72]. A *p*-value < 0.05 was considered statistically significant.

## 3. Results

### 3.1. Participants

A total of 62 participants were initially recruited to the intervention group (Figure 1). One participant was subsequently withdrawn by the healthcare team, resulting in 61 participants completing the initial evaluation. Of these, 50 had an EOS assessment, as 11 participants were not included because of decease (n = 5), self-exclusion (n = 3), loss in follow-up (n = 2), or relapse (n = 1). Data from both the initial and EOS assessments were obtained from 45 participants (72.6% of the recruited participants). For the subgroup analysis, a total of 38 participants (84.4% of those who completed the EOS assessment) were identified as being highly involved in the nutritional intervention and were defined as participants with both high participation (≥4 visits) and high engagement levels, which were determined subjectively by the RDs after one year [47].

In the control group, 85 participants were enrolled (Figure 2), with 8 presenting relapse (n = 1), incomplete treatment (n = 3), inability to collect blood samples (n = 1), or missing dietary data (n = 3), resulting in data analysis for 77 participants (90.6% of the recruited controls).

The socio-demographic characteristics of the participants are presented in Table 1. No differences were found in the characteristics of the control group and the intervention group, including the subgroup with high involvement in the nutritional intervention. While the proportion of males was slightly higher in the intervention group (51.1%) than in the controls (44.2%), the difference was not statistically significant. The mean age at the EOS assessment was 12.0 ± 5.6 y in the intervention group and 10.2 ± 4.5 y in the controls. The EOS assessment occurred 1.3 ± 0.8 y (controls) and 1.4 ± 0.8 y (intervention) after EOT. In all groups, leukemia was the most common diagnosis.

### 3.2. Comparison of Dietary Intakes

The participants’ dietary intakes at the EOS assessment are presented in Table 2. The mean total caloric intake was lower in the intervention group compared to the control group (1759 ± 513 kcal vs. 1997 ± 669 kcal, *p* = 0.042). While the same trend was observed in the subgroup of participants with high involvement, the difference did not reach statistical significance. The percentage of EER was comparable between the groups. Moreover, the mean protein-derived energy was higher in the intervention group compared to the control group, but the difference was statistically different only when compared to highly involved participants (17.2 ± 4.6% vs. 15.3 ± 4.1%, *p* = 0.029). Of note, among the macronutrients, dietary fiber intake was below the AI for 95.6% of the participants in the intervention group, 94.7% in the intervention subgroup, and 93.5% in the control group.

The mean energy-adjusted calcium intake was higher in the intervention group compared to the controls (547.5 ± 240.0 mg/1000 kcal vs. 432.3 ± 197.1 mg/1000 kcal, *p* = 0.005), with an even more pronounced difference in the subgroup with high involvement (569.6 ± 240.3 mg/1000 kcal, *p* = 0.001). Despite this improvement, the proportion of participants who failed to meet the RDA for calcium was not different between the groups. The intervention did not impact vitamin D intake, with only one participant in the control group and two participants of the intervention group meeting the daily recommendation.

Additionally, most of the participants across all the groups had sodium intakes exceeding the UL, with 64.4%, 68.4%, and 55.8% of the participants surpassing this threshold in the intervention group, intervention subgroup, and control group, respectively. The analysis of the data based on age group at diagnosis revealed similar trends (Appendix A). However, in the participants diagnosed at <10 y old, we found that the mean energy-adjusted sodium intake was higher in the intervention group compared to the controls (1438.9 ± 389.9 vs. 1241.9 ± 419.2, *p* = 0.045); this is a difference that was not observed in those diagnosed at an older age. Finally, there was no difference in the mean diet scores DQI and HDI between the groups (Table 3) nor when data were stratified by age group at diagnosis (Appendix A).

### 3.3. Anthropometric Measures

The comparison of anthropometric measures revealed no differences between the control and intervention groups, nor when the controls were compared to the participants highly involved in the nutritional intervention (Table 4) or when the data were stratified based on age group at diagnosis (Appendix A).

### 3.4. Blood Pressure, Biochemical Parameters, and Cardiometabolic Risk Factors

As shown in Table 5, the intervention did not impact the BP and biochemical parameters. However, although no differences met the statistical significance threshold, there was a tendency for better cardiometabolic health in the adolescents who participated in the VIE intervention group.

This is reflected by a proportion of adolescent participants with high BP, nearly half of the control group (21.4% vs. 40.6%; *p* = 0.316), and by the observation that none of the adolescents in the intervention group exhibited high HbA1c levels, whereas three adolescents in the control group did (9.4%; *p* = 0.558) (Table 6). In parallel, the adolescents in the intervention group (n = 14) had lower mean diastolic BP (66 ± 8 vs. 61 ± 8 mmHg; *p* = 0.071) and HbA1c levels (5.19 ± 0.27 vs. 5.01 ± 0.34%; *p* = 0.082) compared to the control group (n = 32), though these differences did not reach the statistical significance (Appendix A).

Furthermore, no association was found between vitamin D status and the study group (Table 5). Nonetheless, with four participants presenting 25(OH)D deficiency levels in each study group, the proportion was twice as high in the intervention group as in the control group without being statistically significant (11.4% vs. 5.4%, *p* = 0.276). Given the endogenous vitamin D production with sunlight exposure [73,74], analyses were carried out to take into account the season in which the assessment was performed for each participant. We found an association between vitamin D status and the season of assessment when data from both groups were analyzed together (Figure 3A), with a higher proportion of participants having vitamin D deficiency during winter, the least sunny period of the year, compared to other seasons (n = 3/8; 37.5%, *p* = 0.018). A similar tendency was observed when analyzing the groups individually; however, statistical significance was not reached. In parallel, we found that more participants of the intervention group had vitamin D levels assessed during winter compared with controls (34.3% vs. 23.4%; *p* = 0.001) (Figure 3B).

## 4. Discussion

This study highlights a modest yet positive impact of the multidisciplinary VIE intervention on the diet of children and adolescents when assessed in the medium term following the completion of cancer treatment (on average 1.3 y). The observed differences were slightly greater in the subgroup of participants who were highly involved in the nutritional intervention. Moreover, while there was a tendency for better cardiometabolic health in the adolescents of the intervention group, the differences did not reach statistical significance.

Following the end of the intervention, the mean caloric intake was lower in the participants in the VIE intervention compared to the controls, but the energy-adjusted intakes of calcium were higher. Even if those intakes showed statistically significant changes, they did not translate to higher diet quality scores (DQI and HDI) in the intervention group compared to the controls. However, the magnitudes of the differences observed between the groups in the specific intakes are, clinically, indicators of better overall dietary quality. The intervention group consumed, on average, 200 fewer kilocalories per day than the control group, which might seem modest, but was rather significant for children and adolescents undergoing growth. As excessive weight gain can often become an important issue during cancer treatment in children and adolescents [75,76], we believed that for this population, if needed, avoiding excessive caloric intake should be preferred to creating an energy deficit. Additionally, the calcium intake, which was higher in the intervention group by a mean of 115 mg/1000 kcal, was clinically significant when put in context with the RDA for children and adolescents ranging between 1000 and 1300 mg [74]. Importantly, calcium intake is a main contributor in the support of the acquisition of peak bone mass, laying the foundation for bone health throughout life.

However, most of the interventions targeting healthy habits in pediatric oncology have been performed only with ALL patients and during the maintenance phase [41,42,43,44,45,46]. Nevertheless, some of these interventions have had positive impacts. Among them, a randomized nutritional counseling intervention of 12 children with ALL, initiated during the maintenance phase, reported a reduction in mean calorie intake after 12 months of follow-up when compared to the baseline, which was not observed in the controls [41]. Similarly, a 12-week lifestyle intervention performed with 15 children with ALL on maintenance therapy and their parents led to an increase in milk consumption and protein intake at the end of the intervention [44]. To our knowledge, only one study other than ours was initiated soon after diagnosis. This pilot study, which included 23 children with ALL, proposed a 6-month dietary intervention initiated within 3 days of starting induction therapy. Comparing dietary intake to the baseline revealed a reduction in sugar intake, in parallel with higher protein and vegetable consumption [77]. These results are in line with those previously described by our group, revealing that, in 36 participants of the VIE intervention, there was an improvement in diet quality, assessed with the DQI and protein intakes at the end of the one-year nutritional intervention, compared with the baseline [49]. With the current study, the main difference is that the dietary outcomes were assessed on average 1.3 years after the completion of the intervention and were compared to a control group. While this limits comparison with the existing literature, it provides a better picture of participants’ ability to maintain, in the medium term, the eating habits acquired during the intervention. The long-term maintenance of the observed differences remains unknown, as it would require an additional study conducted post-intervention with VIE participants in the longer term.

It is worth noting that although the VIE intervention participants had higher energy-adjusted calcium intake than the controls, over half (56.5%) still failed to meet the RDA. This persistent inadequacy underscores the challenge of achieving sufficient calcium intake, even with targeted intervention. Similarly, vitamin D deficiency emerged as a pressing issue, with most of the participants across all the groups not meeting the intake recommendations. Nonetheless, seasonal variability must be considered as a potential bias when comparing vitamin D status among groups. Hence, a larger proportion of participants in the VIE intervention group were assessed during winter and spring, when vitamin D levels are typically lowest in Canada due to decreased UVB-induced endogenous synthesis [73,74]. Concurrently, inadequate calcium and vitamin D intakes are well documented among childhood cancer survivors and are associated with an increased risk of bone complications, including osteoporosis and early fractures [30,31,32,33]. These findings warrant the need for tailored strategies to address persistent calcium and vitamin D deficiencies in pediatric cancer survivors, accounting for both seasonal variability and individual challenges.

Our analysis demonstrated that high involvement in the intervention, as defined by the level of participation and engagement [47], had limited impact on the study outcomes, with only specific higher dietary intakes in calcium and protein compared to the controls. While higher calcium intakes correspond to a significant clinical effect, the greater protein-derived energy intake among the high-involvement participants was only 1.9% higher than the controls, which is a difference with limited clinical impact. Nonetheless, these results are consistent findings across the literature and thus reinforce the need for involvement tracking to contextualize intervention outcomes [78,79]. Integrating assessment strategies to approximate adherence behavior allows an assessment of whether observed changes in health outcomes can be reliably attributed to the intervention [80]. Future interventions should prioritize strategies to enhance involvement to optimize their impact on long-term outcomes.

Weight gain can be an important issue during cancer treatment in children and adolescents. In children undergoing treatment for ALL, BMI tends to increase at the end of the induction phase in relation to corticotherapy [75]. A high BMI or the presence of obesity at diagnosis has been associated with an increased risk of complications [81] (i.e., wound infection and arterial thrombosis), obesity in survivorship [23,82], and, in ALL, disease relapse [18]. Hence, it is important to better understand and address this issue. In our study, we found no difference in the mean of the anthropometric parameters or in the proportion of obesity between the VIE and control groups. However, the prevalence of obesity was quite low in both groups, i.e., 14.3% in the VIE participants and 11.8% in the controls, which may explain why the intervention had no effect on this parameter. In the literature, a nutritional intervention during the maintenance phase led to lower weight gain in patients who took part in the intervention compared to controls, but the measurements were taken immediately after the intervention [46]. Another study, after a median time of 5 years since the end of treatment, observed less weight gain in the intervention group only in adolescents aged over 14 years old [83]. In contrast, and according to our results, other studies reporting the impact of lifestyle interventions to promote weight management in pediatric oncology failed to demonstrate an effect on anthropometric parameters [41,42,43,44,77,84].

While cancer treatments can affect one’s metabolism, including blood lipids and glucose [20,29,85] and nutritional status [86], the role of nutrition in preventing cardiometabolic complications in this acute context is not known. Given the aggressive nature of chemotherapy and radiotherapy in pediatric cancer treatment protocols, their systemic effect might persist for some time after completion, as suggested by the important proportion of metabolic complications observed 1 to 5 years after the end of treatment [25,87,88]. Moreover, as for the general population, it is difficult to clearly demonstrate the impact of preventive interventions on cardiometabolic health in the short term. In our study, the VIE intervention did not significantly impact cardiometabolic health outcomes. Similarly, in a healthy lifestyle intervention with ALL pediatric patients, no difference in TC, HDL-C, triglycerides, and HbA1c, assessed at the end of the intervention, was found between the control and intervention patients [83]. Nonetheless, in the general population, an unbalanced diet is a significant but modifiable risk factor for cardiovascular disease, type 2 diabetes, metabolic syndrome, and osteoporosis [30,31,32,33,35,36,37,38,39,40]. It is possible that the impact of a healthy lifestyle intervention is more observable over the long term after cancer as studies have widely documented the higher prevalence of dyslipidemia, hypertriglyceridemia, high LDL-C, and low HDL-C in older survivors of pediatric cancer [23,29,89,90,91]. Thus, we believe that the better promotion of healthy lifestyle habits in pediatric oncology with medical care teams, patients, and families could have long-term beneficial effects in this population.

Finally, we and others support the importance of age-specific particularities when evaluating intervention effectiveness [78,79,92,93] and that lifestyle interventions must consider adolescents’ specific needs [80] Thus, the analysis with stratification based on age at diagnosis revealed a higher sodium intake in the intervention group only in children. This result has limited clinical significance, as it did not influence the proportion of participants exceeding the UL compared with the controls. Also, in both children and adolescents, this proportion is consistent with what is observed in the Canadian population, where up to 72% of children aged 4 to 13 exceed the UL [94]. Of note, trends (not statistically significant) toward a lesser proportion of cardiometabolic risk factors (high BP and high HbA1c) in the intervention group compared to the controls were found only in adolescents. This absence of statistically significant results is not surprising, considering the group size after stratification (i.e., 46 adolescents and small calculated effect sizes (0.185 and 0.193 for high BP and high HbA1c, respectively). Indeed, a group size of 230 participants would have been necessary to reach 80% power with an alpha of 0.05. Nonetheless, this observation is noteworthy as we have previously shown that, compared to children, older participants were less likely to have a high participation level in the VIE nutritional intervention and that their cardiometabolic health after pediatric cancer treatment was more impacted [25,47]. Taken together, we believe that further studies should specifically focus on tailoring interventions to meet the needs and challenges of adolescents to maximize their participation and the impact of the intervention on cardiometabolic risk factors.

### Strengths and Limitations

Our study has several strengths, including the considerable group size for a study of this nature. In the literature, only two studies on healthy lifestyle intervention during pediatric cancer treatment had a larger or equivalent number of participants [95,96]. Also, the multidisciplinary aspect of the VIE study allowed the participants to receive comprehensive care to maximize the beneficial impact of the intervention. Moreover, we were able to describe a positive impact of the intervention on diet 1.3 years after the end of the intervention; this is a unique aspect allowing the assessment of the persistence of the effect of our approach. The presence of a control group that was comparable in terms of age, sex, and diagnosis to the intervention group further strengthens our assessment of the intervention impact. Finally, our study included several cancer diagnoses, making the results more generalizable and representative for this population.

Our study also has limitations. Of note, the recruitment method could introduce biases as it was dependent on whether the participants were treated before or after the initiation of the VIE intervention at the CHUSJ and was also based on a convenience sample and voluntary participation. Hence, it is possible that the recruited participants were more aware of the importance of diet and healthy lifestyle habits and therefore more inclined to have a better-quality diet, which could explain the differences observed between the two groups. Moreover, the COVID−19 pandemic occurred during the VIE intervention, while the assessments of the control group had been performed beforehand. The stress and isolation experienced by families may have influenced their lifestyle habits and, consequently, the outcomes at the end-of-study assessment. Moreover, for feasibility and logistical reasons, a single 24-h recall was used to collect the participants’ food intake. It has been shown that the 3-day food record or repetitive 24-h recalls are more reliable to assess usual food intake of individuals [97,98]. Additionally, as much as the heterogeneity of diagnoses is a study strength, this implies a high degree of variability in the participants’ treatment protocols. Finally, factors such as health status and lifestyle habits before cancer diagnosis, as well as psychosocial aspects, were not considered in this study, but they could have influenced the cardiometabolic health and diet quality of the participants in both groups.

## 5. Conclusions

In conclusion, the VIE multidisciplinary intervention had a slight yet positive impact on caloric and calcium intakes without translating to better diet quality scores in the medium term after treatment completion when compared to a control group. In the adolescents only, the intervention group exhibited a trend toward a lower proportion of cardiometabolic risk factors at the time of assessment. Although these findings showed limited clinical impact 1.3 years after treatment, they underscore the importance of tailoring early healthy lifestyle interventions, especially for adolescents, and exploring additional strategies to improve the diet of all pediatric cancer patients during and after treatment. Further research is needed to assess the long-term impact of such interventions.

## Figures and Tables

**Figure 1 cancers-17-00157-f001:**
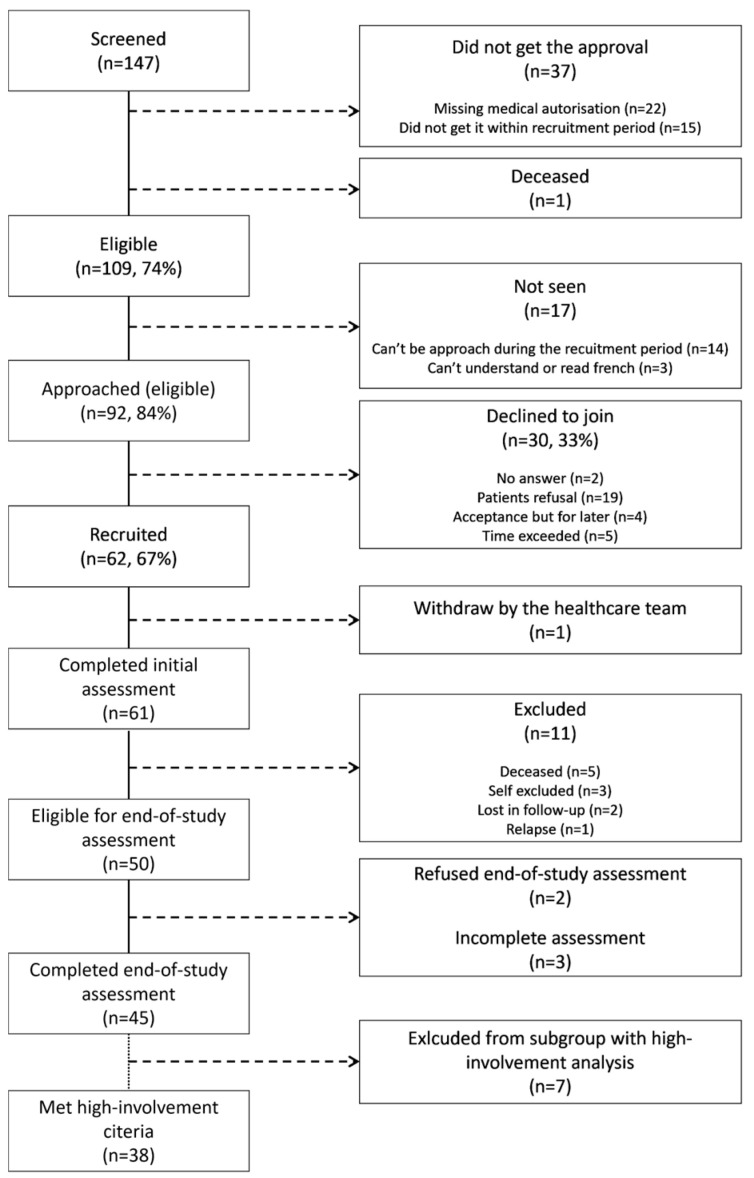
Flow diagram of the intervention group and subgroup.

**Figure 2 cancers-17-00157-f002:**
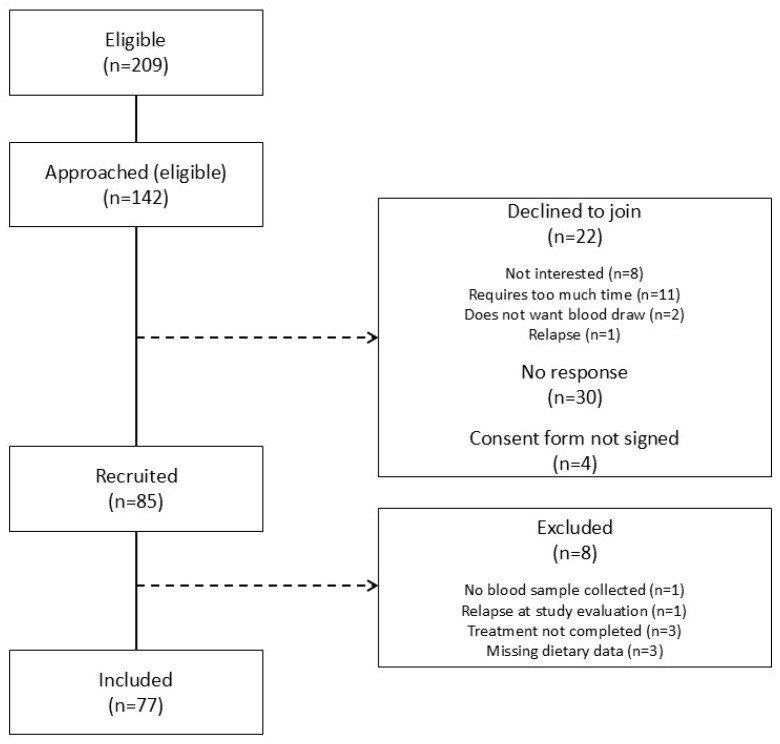
Flow diagram of the control group.

**Figure 3 cancers-17-00157-f003:**
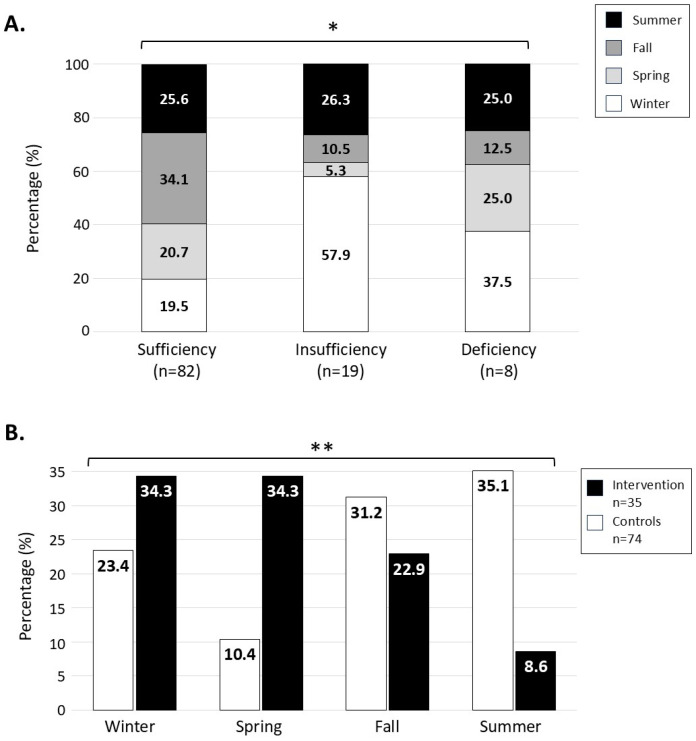
Vitamin D status according to seasons in the intervention group. 25-hydroxyvitamin D [25(OH)D] was measured in serum using liquid chromatography–tandem mass spectrometry. The relationship between season (summer vs. fall vs. spring vs. winter) of sampling and (**A**) vitamin D status (sufficiency vs. insufficiency vs. deficiency) and (**B**) study group (control vs. VIE intervention) was assessed using Fisher’s exact test and chi-square test, respectively. * *p* < 0.05; ** *p* < 0.01.

**Table 1 cancers-17-00157-t001:** Characteristics of participants.

	Control Group n = 77	Intervention Group n = 45	*p*-Value ^1^	Intervention Subgroup ^2^ n = 38	*p*-Value ^3^
**Sex, n (%)**					
Male	34 (44.2)	23 (51.1)	0.573	21 (55.3)	0.356
Female	43 (55.8)	22 (48.9)		17 (44.7)	
**Age at diagnosis (y)**					
Mean ± SD	9.1 ± 5.6	7.5 ± 4.7	0.106	7.8 ± 4.8	0.205
Median (min–max)	7.4 (1.1–17.9)	5.7 (1.3–16.1)		5.8 (1.3–16.1)	
**Age at EOT (y)**					
Mean ± SD	10.6 ± 5.5	8.9 ± 4.5	0.077	9.2 ± 4.7	0.162
Median (min–max)	7.9 (2.8–19.7)	7.0 (1.9–18.1)		7.1 (1.9–18.1)	
**Age at EOS (y)**					
Mean ± SD	12.0 ± 5.6	10.2 ± 4.5	0.064	10.4 ± 4.7	0.134
Median (min–max)	9.9 (4.5–21.0)	8.3 (3.2–18.9)		8.6 (3.2–18.9)	
**Age categories at EOS, n (%)**					
Children (<10 y)	44 (57.1)	30 (66.7)	0.340	24 (63.2)	0.678
Adolescents (≥10 y)	33 (42.9)	15 (33.3)		14 (36.8)	
**Time between EOT and EOS (y)**					
Mean ± SD	1.4 ± 0.8	1.3 ± 0.8	0.465	1.3 ± 0.8	0.422
Median (min–max)	1.4 (0.0–3.5)	1.3 (0.1–2.7)		1.2 (0.1–2.7)	
**Cancer diagnosis, n (%)**					
Leukemia ^4^	33 (42.9)	25 (55.6)	0.507	22 (57.9)	0.340
Lymphoma ^5^	19 (24.7)	7 (15.6)		6 (15.8)	
Sarcoma ^6^	8 (10.4)	5 (11.1)		5 (13.2)	
Other ^7^	17 (22.1)	8 (17.8)		5 (13.2)	
**Treatment duration (y)**					
Mean ± SD	1.5 ± 0.8	1.4 ± 0.8	0.461	1.4 ± 0.8	0.534
Median (min–max)	2.1 (0.2–3.2)	1.4 (0.2–2.4)		1.5 (0.2–2.4)	

Comparison between the control and intervention group or subgroup was conducted using Pearson chi-square test (categorical variables) and Student’s *t*-test (continuous variables). ^1^
*p*-value for the comparison between the control group and intervention group. ^2^ High-involvement intervention subgroup was defined as participants with both high participation (≥4 visits) and high engagement. Engagement levels were assessed subjectively by the RDs after one year of intervention and categorized as low, moderate, or high [47]. ^3^
*p*-value for the comparison between the control group and intervention subgroup.^4^ Leukemia diagnosis includes acute lymphoblastic leukemia, B mature leukemia with mixed-lineage leukemia, and acute myeloid leukemia. ^5^ Lymphoma diagnosis includes Hodgkin’s lymphoma, Burkitt’s lymphoma, anaplastic lymphoma, primary central nervous system lymphoma, and lymphoblastic lymphoma. ^6^ Sarcoma diagnosis includes osteosarcoma, rhabdomyosarcoma, synovial sarcoma, and Ewing’s sarcoma. ^7^ Other diagnoses include Wilm’s tumor, germinoma, mixed brain germ cell tumor, medulloblastoma, neuroblastoma, and hepatoblastoma. SD: standard deviation; y: years; BMI: body mass index.

**Table 2 cancers-17-00157-t002:** Comparison of dietary intakes at end-of-study assessment.

	Control Group n = 77	Intervention Group n = 45	*p*-Value ^1^	Intervention Subgroup ^2^ n = 38	*p*-Value ^3^
**Energy**					
kcal					
Mean ± SD	1997 ± 669	1759 ± 513	0.042	1803 ± 526	0.121
Median (min–max)	1854 (810–4370)	1695 (1007–3351)		1720 (1074–3351)	
kcal/kg					
Mean ± SD	58.0 ± 31.4	57.4 ± 28.9	0.917	56.9 ± 28.9	0.861
Median (min–max)	51.7 (15.3–160.7)	54.6 (14.2–116.5)		53.0 (14.2–116.5)	
% of EER					
Mean ± SD	99.4 ± 26	94.8 ± 25.7	0.342	95.2 ± 25.7	0.417
Median (min–max)	100.4 (49.0–164.5)	94.8 (41.0–141.6)		95.0 (41.0–141.6)	
**Fat**					
g/kg					
Mean ± SD	2.1 ± 1.3	2.3 ± 1.3	0.578	2.3 ± 1.3	0.609
Median (min–max)	1.7 (0.3–5.8)	2.0 (0.4–5.5)		2.0 (0.4–5.5)	
%Energy					
Mean ± SD	32.6 ± 7.7	35.4 ± 7.8	0.055	35.3 ± 8.1	0.082
Median (min–max)	31.8 (18.5–49.4)	35.2 (14.9–52.2)		35.3 (14.9–52.2)	
n (%)					
>AMDR	26 (33.8)	23 (51.1)	0.057	20 (52.6)	0.083
=AMDR	38 (49.4)	20 (44.4)		16 (42.1)	
<AMDR	13 (16.9)	2 (4.4)		2 (5.3)	
**Carbohydrates**					
g/kg					
Mean ± SD	7.6 ± 4.5	7.1 ± 3.7	0.521	6.9 ± 3.6	0.417
Median (min–max)	6.7 (1.9–25.1)	7.3 (1.5–15.8)		6.7 (1.5–15.8)	
%Energy					
Mean ± SD	52.5 ± 9.7	49.4 ± 9.4	0.089	48.9 ± 9.9	0.064
Median (min–max)	53.3 (22.0–72.9)	49.0 (27.1–70.5)		48.7 (27.1–70.5)	
n (%)					
>AMDR	7 (9.1)	2 (4.4)	0.320	2 (4.4)	0.171
=AMDR	54 (70.1)	29 (64.4)		22 (64.4)	
<AMDR	16 (20.8)	14 (31.1)		14 (31.1)	
**Proteins**					
g/kg					
Mean ± SD	2.2 ± 1.1	2.4 ± 1.4	0.370	2.4 ± 1.4	0.286
Median (min–max)	2.1 (0.4–4.8)	1.9 (0.4–6.0)		2.1 (0.4–6.0)	
%Energy					
Mean ± SD	15.3 ± 4.1	16.6 ± 4.7	0.121	17.2 ± 4.6	0.029
Median (min–max)	14.9 (8.7–32.6)	16.2 (8.8–27.0)		17.1 (10.1–27.0)	
%RDA					
Mean ± SD	236.4 ± 116.3	251.2 ± 144.1	0.535	257.0 ± 143.6	0.411
Median (min–max)	228.7 (48.9–506.6)	208.5 (52.0–607.2)		217.6 (52.0–607.2)	
n (%)					
>RDA	71 (92.2)	41 (91.1)	0.541	36 (94.7)	0.616
<RDA	6 (7.8)	4 (8.9)		2 (5.3)	
**Dietary fiber**					
g/1000 kcal					
Mean ± SD	8.4 ± 2.8	9.2 ± 3.4	0.188	9.0 ± 3.1	0.322
Median (min–max)	8.3 (3.9–19.5)	8.9 (3.7–19.3)		8.6 (3.7–15.6)	
%AI					
Mean ± SD	58.1 ± 22.7	60.6 ± 27.8	0.586	60.5 ± 27.6	0.618
Median (min–max)	56.0 (18.8–127.5)	57.9 (17.2–112.6)		56.6 (17.2–112.6)	
n (%)					
>AI	5 (6.5)	2 (4.4)	0.713	2 (5.3)	0.795
<AI	72 (93.5)	43 (95.6)		36 (94.7)	
**Sodium**					
mg/1000 kcal					
Mean ± SD	1358.2 ± 546.5	1414.5 ± 397.4	0.548	1438.9 ± 404.9	0.421
Median (min–max)	1312.9 (446.9–2920.5)	1412.8 (595.3–2330.4)		1407.2 (653.4–2330.4)	
%UL					
Mean ± SD	128.6 ± 68.8	122.6 ± 48.6	0.610	126.8 ± 48.9	0.883
Median (min–max)	105.0 (34.8–311.1)	108.5 (26.1–237.3)		110.9 (59.6–237.3)	
n (%)					
>UL	43 (55.8)	29 (64.4)	0.446	26 (68.4)	0.228
<UL	34 (44.2)	16 (35.6)		12 (31.6)	
**Calcium**					
mg/1000 kcal					
Mean ± SD	432.3 ± 197.1	547.5 ± 240.0	0.005	569.6 ± 240.3	0.001
Median (min–max)	390.3 (127.4–1218.9)	523.3 (124.0–1206.6)		536.8 (194.0–1206.6)	
%RDA					
Mean ± SD	81.3 ± 49.1	89.1 ± 48.3	0.393	94.0 ± 48.7	0.195
Median (min–max)	70.9 (13.4–260.6)	86.6 (9.6–209.8)		92.1 (26.5–209.8)	
n (%)					
>RDA	28 (36.4)	20 (43.5)	0.451	18 (47.4)	0.313
<RDA	49 (63.6)	26 (56.5)		20 (52.6)	
**Vitamin C**					
mg/1000 kcal					
Mean ± SD	59.6 ± 44.1	59.9 ± 47.0	0.975	55.2 ± 42.9	0.610
Median (min–max)	52.5 (0.8–183.0)	46.4 (1.8–191.0)		46.4 (1.8–141.1)	
%RDA					
Mean ± SD	295.9 ± 250.3	338.6 ± 330.5	0.421	322.7 ± 327.5	0.627
Median (min–max)	235.1 (1.9–1123.5)	206.1 (7.6–1156.2)		184.3 (7.6–1156.2)	
n (%)					
>RDA	58 (75.3)	32 (71.1)	0.672	26 (68.4)	0.504
<RDA	19 (24.7)	13 (28.9)		12 (31.6)	
**Vitamin D**					
µg/1000 kcal					
Mean ± SD	2.2 ± 2.0	2.6 ± 2.0	0.311	2.9 ± 2.1	0.095
Median (min–max)	1.9 (0.0–9.9)	2.2 (0.1–9.3)		2.6 (0.1–9.3)	
%RDA					
Mean ± SD	29.6 ± 24.8	31.8 ± 27.8	0.646	35.7 ± 28.3	0.235
Median (min–max)	26.6 (0.2–101.3)	25.2 (1.3–119.8)		34.2 (1.3–119.8)	
n (%)					
>RDA	1 (1.3)	2 (4.4)	0.554	2 (5.3)	0.253
<RDA	76 (98.7)	43 (95.6)		36 (94.7)	

Dietary data were collected using a 24-h recall at the end-of-study (EOS) assessment. Comparison between the control and the intervention group or subgroup was conducted using Pearson chi-square test (categorical variables) and Student’s *t*-test (continuous variables). ^1^
*p*-value for the comparison between the control group and intervention group. ^2^ High-involvement intervention subgroup was defined as participants with both high participation (≥4 visits) and high engagement. Engagement levels were assessed subjectively by the RDs after one year of intervention and categorized as low, moderate, or high [47]. ^3^
*p*-value for the comparison between the control group and intervention subgroup. SD: standard deviation; EER: estimated energy requirement; AMDR: acceptable macronutrient distribution range; RDA: recommended dietary allowance; AI: adequate intake; UL: tolerable upper intake level.

**Table 3 cancers-17-00157-t003:** Comparison of diet quality scores at end-of-study assessment.

	Control Group n = 77	Intervention Group n = 45	*p*-Value ^1^	Intervention Subgroup ^2^ n = 38	*p*-Value ^3^
**Diet Quality Index (DQI) ^4^**					
Mean ± SD	52.8 ± 12.9	51.8 ± 11.6	0.694	52.3 ± 11.3	0.845
Median (min–max)	53.5 (27.0–74.0)	54.5 (30.5–77.5)		55.2 (32.5–77.5)	
**Healthy Diet Indicator (HDI) ^5^**					
Mean ± SD	3.0 ± 1.4	2.6 ± 1.2	0.159	2.8 ± 1.1	0.463
Median (min–max)	3.0 (0–7)	2.0 (0–6)		3.0 (1–6)	
Level of adherence, n (%)					
Low	51 (66.2)	35 (77.8)	0.350	29 (76.3)	0.452
Medium	24 (31.2)	10 (22.2)		9 (23.7)	
High	2 (2.6)	0 (0)		0 (0)	

Dietary quality scores were calculated from nutritional data collected using 24-h recall at the end-of-study (EOS) assessment. Comparison between the control group and the intervention group or subgroup was conducted using Student’s *t*-test and Fisher’s exact test (level of adherence). ^1^
*p*-value for the comparison between the control group and intervention group. ^2^ High-involvement intervention subgroup was defined as participants with both high participation (≥4 visits) and high engagement. Engagement levels were assessed subjectively by the RDs after one year of intervention and categorized as low, moderate, or high [47]. ^3^
*p*-value for the comparison between the control group and intervention subgroup. ^4^ DQI is a continuous score with a maximum of 100 points, where a higher total score indicates a better quality diet [57]. ^5^ HDI is based on the sum of 9 components for which a value of +1 is assigned when intake is within the recommended range or otherwise as 0, generating a score ranging from 0 to 9. The level of adherence is determined as low (0–3), medium (4–6), and high (≥7) [58,59,60]. SD: standard deviation.

**Table 4 cancers-17-00157-t004:** Comparison of anthropometric measures at end-of-study assessment.

	Control Group	Intervention Group	*p*-Value ^1^	Intervention Subgroup ^2^	*p*-Value ^3^
**Weight, z-score**	n = 77	n = 45		n = 38	
Mean ± SD	0.28 ± 1.17	0.45 ± 1.48	0.482	0.51 ± 1.55	0.358
Median (min–max)	0.18 (−1.78–3.59)	0.16 (−1.98–6.18)		0.18 (−1.98–6.18)	
**Height, z-score**	n = 76	n = 45		n = 38	
Mean ± SD	−0.08 ± 0.94	−0.25 ± 0.94	0.353	−0.19 ± 0.96	0.569
Median (min–max)	−0.09 (−2.40–2.78)	−0.09 (−3.26–2.17)		−0.07 (−3.26–2.17)	
**BMI, z-score**	n = 76	n = 45		n = 38	
Mean ± SD	0.38 ± 1.20	0.71 ± 1.36	0.167	0.74 ± 1.38	0.149
Median (min–max)	0.27 (−2.09–3.43)	0.48 (−1.68–5.62)		0.56 (−1.68–5.62)	
**WC, z-score**	n = 69	n = 39		n = 32	
Mean ± SD	0.73 ± 0.69	0.64 ± 0.97	0.584	0.65 ± 0.98	0.632
Median (min–max)	0.74 (−1.19–2.22)	0.71 (−1.79–2.15)		0.74 (−1.79–2.15)	
**MUAC, z-score**	n = 75	n = 42		n = 35	
Mean ± SD	0.80 ± 0.91	0.49 ± 1.35	0.151	0.48 ± 1.42	0.159
Median (min–max)	0.75 (−1.12–3.05)	0.53 (−4.95–2.61)		0.53 (−4.95–2.61)	
**TSFT, z-score**	n = 71	n = 41		n = 34	
Mean ± SD	0.35 ± 1.36	0.39 ± 0.90	0.863	0.42 ± 0.92	0.768
Median (min–max)	0.43 (−2.66–3.47)	0.45 (−1.73–2.23)		0.56 (−1.73–2.23)	
**SSFT, z-score**	n = 68	n = 39		n = 33	
Mean ± SD	0.12 ± 1.14	0.18 ± 1.11	0.806	0.17 ± 1.20	0.854
Median (min–max)	−0.02 (−2.79–2.70)	0.17 (−2.64–2.30)		0.15 (−2.64–2.30)	

Data on anthropometric measures were collected at the end-of-study (EOS) assessment. Comparison between the control group and the intervention group or subgroup was conducted using Student’s *t*-test. ^1^
*p*-value for the comparison between the control group and intervention group. ^2^ High-involvement intervention subgroup was defined as participants with both high participation (≥4 visits) and high engagement. Engagement levels were assessed subjectively by the RDs after one year of intervention and categorized as low, moderate, or high [47]. ^3^
*p*-value for the comparison between the control group and intervention subgroup. BMI: body mass index; MUAC: mid-upper arm circumference; TSFT: triceps skinfold thickness; SSFT: subscapular skinfold thickness; SD: standard deviation; WC: waist circumference.

**Table 5 cancers-17-00157-t005:** Comparison of blood pressure and biochemical parameters at end-of-study assessment.

	Control Group	Intervention Group	*p*-Value ^1^	Intervention Subgroup ^2^	*p*-Value ^3^
**Blood pressure**					
Systolic, mmHg	n = 76	n = 42		n = 35	
Mean ± SD	104 ± 14	103 ± 10	0.845	104 ± 10	0.899
Median (min–max)	101 (73–136)	101 (88–134)		101 (88–134)	
Systolic, z-score	n = 76	n = 39		n = 32	
Mean ± SD	0.01 ± 1.00	0.01 ± 0.79	0.992	0.04 ± 0.831	0.976
Median (min–max)	0.07 (−2.33–2.33)	−0.09 (−1.41–1.69)		−0.10 (−1.41–1.69)	
Diastolic, mmHg	n = 76	n = 42		n = 35	
Mean ± SD	61 ± 8	61 ± 10	0.983	60 ± 8	0.737
Median (min–max)	60 (42–84)	59 (45–98)		59 (45–75)	
Diastolic, z-score	n = 76	n = 39		n = 32	
Mean ± SD	−0.07 ± 0.64	0.05 ± 1.01	0.419	−0.03 ± 0.852	0.791
Median (min–max)	−0.08 (−1.75–1.75)	−0.08 (−2.24–3.92)		−0.13 (−2.24–1.84)	
**Blood lipids**					
TC, mmol/L	n = 77	n = 42		n = 35	
Mean ± SD	4.15 ± 0.82	4.21 ± 1.14	0.720	4.26 ± 1.24	0.566
Median (min–max)	4.05 (2.27–6.79)	4.01 (2.29–8.53)		3.99 (2.29–8.53)	
TC classification, n (%)	n = 77	n = 42		n = 35	
Normal value	68 (88.3)	37 (88.1)	1.000	30 (85.7)	0.939
High value	9 (11.7)	5 (11.9)		5 (14.3)	
LDL-C, mmol/L	n = 77	n = 41		n = 34	
Mean ± SD	2.46 ± 0.74	2.43 ± 1.13	0.862	2.50 ± 1.22	0.837
Median (min–max)	2.39 (0.60–5.04)	2.24 (0.84–6.80)		2.26 (0.84–6.80)	
LDL-C classification, n (%)	n = 77	n = 41		n = 34	
Normal value	68 (88.3)	37 (90.2)	0.773	30 (88.2)	1.000
High value	9 (11.7)	4 (9.8)		4 (11.8)	
HDL-C, mmol/L	n = 77	n = 41		n = 34	
Mean ± SD	1.30 ± 0.28	1.31 ± 0.31	0.908	1.28 ± 0.30	0.785
Median (min–max)	1.31 (0.68–2.01)	1.28 (0.74–1.85)		1.27 (0.77–1.85)	
HDL-C classification, n (%)	n = 77	n = 41		n = 34	
Normal value	62 (80.5)	33 (80.5)	1.000	27 (79.4)	1.000
Low value	15 (19.5)	8 (19.5)		7 (20.6)	
**Glucose metabolism**					
HbA1c (%)	n = 71	n = 33		n = 27	
Mean ± SD	5.15 ± 0.29	5.12 ± 0.36	0.705	5.06 ± 0.33	0.230
Median (min–max)	5.10 (4.30–6.00)	5.20 (4.40–5.80)		5.20 (4.40–5.70)	
**Micronutrient status**					
Vitamin D (nmol/L)	n = 74	n = 35		n = 30	
Mean ± SD	70.2 ± 23.5	65.4 ± 29.3	0.360	69.1 ± 29.0	0.845
Median (min–max)	69.5 (15.0–132.8)	61.3 (14.5–151.3)		63.3 (14.5–151.3)	
Vitamin D classification, n (%)	n = 74	n = 35		n = 30	
Sufficiency	59 (79.7)	23 (65.7)	0.276	22 (73.3)	0.773
Insufficiency	11 (14.9)	8 (22.9)		6 (20.0)	
Deficiency	4 (5.4)	4 (11.4)		2 (6.7)	

Data on cardiometabolic health indicators were collected at the end-of-study (EOS) assessment. Comparison between the control group and the intervention group or subgroup was conducted using Pearson chi-square test (categorical variables) and Student’s *t*-test (continuous variables). ^1^
*p*-value for the comparison between the control group and intervention group. ^2^ High-involvement intervention subgroup was defined as participants with both high participation (≥4 visits) and high engagement. Engagement levels were assessed subjectively by the RDs after one year of intervention and categorized as low, moderate, or high [47]. ^3^
*p*-value for the comparison between the control group and intervention subgroup. HbA1C: glycosylated hemoglobin; TC: total cholesterol; HDL-C: high-density lipoprotein cholesterol; LDL-C: low-density lipoprotein cholesterol; SD: standard deviation.

**Table 6 cancers-17-00157-t006:** Comparison of the prevalence of cardiometabolic risk factors.

	Control Group	Intervention Group	*p*-Value ^1^	Intervention Subgroup ^2^	*p*-Value ^3^
**High blood pressure, n (%)**					
All participants	n = 76	n = 42		n = 35	
	20 (26.3)	9 (21.4)	0.658	8 (22.9)	0.877
Children ^4^	n = 44	n = 28		n = 22	
	7 (15.9)	6 (21.4)	0.754	5 (22.7)	0.735
Adolescents ^5^	n = 32	n = 14		n = 13	
	13 (40.6)	3 (21.4)	0.316	3 (23.1)	0.441
**High HbA1c, n (%)**					
All participants	n = 72	n = 33		n = 27	
	5 (6.9)	2 (6.1)	1.000	1 (3.7)	0.897
Children ^4^	n = 40	n = 22		n = 17	
	2 (5.0)	2 (9.1)	0.610	1 (5.9)	1.000
Adolescents ^5^	n = 32	n = 11		n = 10	
	3 (9.4)	0 (0.0)	0.558	0 (0.0)	0.763
**Blood lipid abnormalities, n (%)**					
All participants	n = 77	n = 42		n = 35	
	25 (32.5)	13 (31.0)	1.000	12 (34.3)	1.000
Children ^4^	n = 44	n = 28		n = 22	
	6 (13.6)	6 (21.4)	0.519	5 (22.7)	0.559
Adolescents ^5^	n = 33	n = 14		n = 13	
	19 (57.6)	7 (50.0)	0.752	7 (53.8)	1.000
**Obesity, n (%)**					
All participants	n = 76	n = 42		n = 35	
	9 (11.8)	6 (14.3)	0.776	6 (17.1)	0.645
Children ^4^	n = 43	n = 28		n = 22	
	3 (7.0)	2 (7.1)	1.000	2 (9.1)	1.000
Adolescents ^5^	n = 33	n = 14		n = 13	
	6 (18.2)	4 (28.6)	0.456	4 (30.8)	0.593
**≥2 risk factors, n (%)**					
All participants	n = 77	n = 42		n = 35	
	14 (18.2)	8 (19.0)	1.000	7 (20.0)	0.894
Children ^4^	n = 44	n = 28		n = 22	
	1 (2.3)	3 (10.7)	0.292	3 (13.6)	0.060
Adolescents ^5^	n = 33	n = 14		n = 13	
	13 (39.4)	5 (35.7)	1.000	4 (30.8)	0.836

Data on the presence of cardiometabolic risk factors were collected at the end-of-study (EOS) assessment. Comparison between the control group and the intervention group or subgroup was conducted using Pearson chi-square test. ^1^
*p*-value for the comparison between the control group and intervention group. ^2^ High-involvement intervention subgroup was defined as participants with both high participation (≥4 visits) and high engagement. Engagement levels were assessed subjectively by the RDs after one year of intervention and categorized as low, moderate, or high [47]. ^3^
*p*-value for the comparison between the control group and intervention subgroup.^4^ Participants aged <10 years old at cancer diagnosis. ^5^ Participants aged ≥10 years old at cancer diagnosis. SD: standard deviation.

## Data Availability

The datasets used and/or analyzed are available from the corresponding author upon reasonable request.

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
