# Peer review of "Impact of Early Nutritional Intervention During Cancer Treatment on Dietary Intakes and Cardiometabolic Health in Children and Adolescents"

_cancers, 2025, doi:10.3390/cancers17010157_

Round 1
Reviewer 1 Report
Comments and Suggestions for Authors
See attached file for detailed comments.

Reviewer 2 Report
Comments and Suggestions for Authors
Dear Author, I give you the following comment. Please address this in your manuscript to enhance the readability and understanding of your manuscript.
Major Comments:
- The study evaluates the impact of the VIE intervention on pediatric cancer survivors, yet the cardiometabolic outcomes did not show statistical significance. Could the authors discuss potential reasons why this intervention might not have had a more pronounced effect on cardiometabolic health, and how future studies could address this gap?
- The study reveals that participants highly involved in the nutritional intervention had greater protein-derived energy intake compared to the control group. How do the authors propose to ensure that these improvements are sustainable over time, especially considering the need for long-term follow-up?
- Given that the intervention group showed improvements in dietary intake but no significant effects on cardiometabolic health, would the authors recommend a different intervention approach or modifications to the current one? If so, what would those changes be?
- The results indicate that there were no differences based on age at diagnosis. Could the authors explore potential mechanisms behind this lack of effect and whether certain subgroups (e.g., younger children vs. adolescents) might benefit from tailored interventions?
- While the study focuses on dietary intake and cardiometabolic health, other factors such as psychosocial aspects of recovery were also part of the intervention. Could the authors elaborate on how these other components might have contributed to the overall results, and whether they may be more influential than dietary factors alone?
Minor Comments:
- The authors mention statistical differences in dietary intake between the intervention and control groups. Could they provide further clarification on whether these differences are clinically significant in the context of pediatric cancer recovery?
- It would be helpful for the authors to include more information about the baseline characteristics of the participants, such as their nutritional status or pre-treatment health conditions, to better understand how these factors may have influenced the outcomes.
- In the discussion, the authors note that further research is warranted to assess the long-term impact of such interventions. Could the authors suggest potential methods for assessing long-term health outcomes in future studies, particularly focusing on cardiometabolic health?
- The abstract mentions statistical differences in caloric and calcium intake between the two groups, but it would be helpful to clarify the overall impact of these changes on participants’ health outcomes, particularly in terms of long-term effects on weight management and bone health.
- The study design mentions stratifying by age at diagnosis, yet the results do not show any significant differences based on this variable. Could the authors clarify whether age at diagnosis was a significant determinant of the success of the intervention, or if other factors, such as adherence to the intervention, played a larger role?
These questions aim to address both overarching concerns and specific technical details that could impact the robustness and clarity of the study's findings.
Best Regards
Comments on the Quality of English LanguageFine
Reviewer 3 Report
Comments and Suggestions for Authors
This is a careful study; its relevance is clear and method and results are meticulously reported. A wide range of indicators of the quality of dietary intake is used.
The only issue is that the conclusion seems too positive to me. I am not convinced that "the VIE multi-disciplinary intervention had a slight yet positive impact on specific dietary intakes in the medium-term after treatment completion when compared to a control group." (Conclusion.) None of these positive differences was statistically significant, there are also several aspects of the diet where the intervention group scored worse (e.g. fat intake), and the scores on the overall measures Diet Quality Index and Healthy Diet Indicator were lower for the intervention group compared to the control group. So I would describe the results as mixed, with no clear positive effect for the intervention.
So I do not think that "these findings support the relevance of early healthy lifestyle intervention in pediatric oncology," at least not following the VIE program. This is of course disappointing, but yet a negative result is also an important result. In my view, the recommendation should be that other ways should be found to improve the diet of children and adolescents during and after cancer treatment.
Round 2
Reviewer 1 Report
Comments and Suggestions for Authors
See attached file for my remaining notes.

Reviewer 3 Report
Comments and Suggestions for Authors
The revision made this high-quality manuscript even better. My comments were adequately processed. No further comments.
Author Response
Comment : The revision made this high-quality manuscript even better. My comments were adequately processed. No further comments.
Response : We would like to thank the Reviewer for his/her comment and for taking the time to review this manuscript.